



# Aircraft measurements of black carbon in the boundary layer over the North China Plain

Delong Zhao[1,2], Mengyu Huang[1,2], Dantong Liu[3,2*], Deping Ding[1,2*], Ping Tian[1,2], Quan Liu[1,2], Wei Zhou[1,2], Jiujiang Sheng[1,2], Fei Wang[1,2], Kai Bi[1,2], Yan Yang[1,2], Xia Li[1,2], Yaqiong Hu[1,2], Xin Guo[1,2], Yang Gao[4], Hui He[1,2],Yunbo Chen[1,2], Shaofei Kong[5], Jiayi Huang[6]

[1]Beijing Weather Modification Office, Beijing, 100089, China

[2]Beijing Key Laboratory of Cloud, Precipitation and Atmospheric Water Resources, Beijing, 100089, China

[3]Centre for Atmospheric Sciences, School of Earth and Environmental Sciences, University of Manchester, Manchester M13 9PL, UK.

[4]Chinese Academy of Meteorological Sciences, Beijing, 100081, China

[5]Department of Atmospheric Sciences, School of Environmental Studies, China University of Geosciences (Wuhan), 430074, Wuhan, China

[6]Nanjing University, 210023, Nanjing, China

Correspondence to: Dantong Liu (dantong.liu@manchester.ac.uk) or Deping Ding (zytddp@vip.sina.com)



## Abstract

Black carbon aerosol (BC) is the principle absorber to modify the shortwave radiative balance between the lower atmosphere and the surface in polluted environment, which was in-situ characterized by aircraft measurements using a single particle soot photometer (DMT inc., SP2) throughout the boundary layer up to 3km over the North China Plain around Beijing megacity. The flights were conducted in both hot season (late spring, Apr. – Jun. 2012, surface temperature >20°C, 10 flights) and cold season (winter, Dec. 2016, surface temperature <5°C, 6 flights). The BC mass in the daytime well-developed planetary boundary layer ($BC_{PBL}$) was found to be largely influenced by meteorology which modulated the local emission and regional transport. The $BC_{PBL}$ in hot season showed no apparent vertical gradient or correlation with the PBL height (PBLH); whereas in winter the $BC_{PBL}$ decreased at higher altitude and accumulated towards the surface due to reduced horizontal wind speed at low level, and was anti-correlated with the PBLH. The more homogenous vertical mixing of $BC_{PBL}$ in hot season may result from the stronger convective mixing due to higher surface temperature, in addition the predominant southerly wind may have advected the largely polluted air mass, offsetting its dilution effect; whereas in winter the predominant cleaner air mass from the north had important dilution effect on local emissions. The overall averages showed lower BC mass loading in the surface layer during hot season but similar loadings (3.5-4 µg m$^{-3}$) in the PBL for both seasons. The highly turbulent air conditions, as frequently observed in the early afternoon during hot season, largely diluted the BC mass down to the background level (~0.5 µg m$^{-3}$); whereas in winter the $BC_{PBL}$ was significantly elevated under dynamic air condition due to efficient transport of polluted air mass from the south.

The $BC_{PBL}$ in winter showed systematically larger core size (mass median diameter, MMD=217±4nm) than in late spring (209±7nm), which may be due to a higher contribution from the residential emission sector in cold season. The high BC mass loadings were mostly associated with a high fraction of thick coatings ($F_{coating}$), and the higher air moisture enhanced the coating content of $BC_{PBL}$. These suggest the primary sources with a range of co-emitted species significantly contributed to the coatings of BC, and the secondary formation which could be enhanced by elevated RH also played important role. The BC after significant removal showed much lower $F_{coating}$ (~0.06) and smaller core size (MMD ~193nm), implying that the coated and larger BC particle was preferentially removed. The scavenging efficiency of BC in the entrainment zone (EZ) showed positive correlation with the change of coatings between PBL and EZ for both seasons ($F_{coating,PBL} > F_{coating,EZ}$), however showed seasonally different behaviors in terms of core size (MMD$_{PBL}$>MMD$_{EZ}$ in late spring, but the opposite in winter). These results provide the basis to evaluate the BC direct radiative forcing in the polluted planetary boundary layer over this region and also the indirect forcing of BC by interacting with low-level clouds.




## 1 Introduction

Black carbon aerosol (BC), as the strong absorber in the visible and infrared, has great regional impact in the atmosphere especially at the polluted hotspots such as in South and East Asia (Ramanathan and Carmichael, 2008). The BC has severe impact on both climate (Bond et al., 2013) and human health (Baumgartner et al.,

2014) in these regions with high anthropogenic emissions and population exposure. Reducing BC has been postulated as a win-win policy intervention because of its shorter atmospheric lifetime compared to the greenhouse gases, which will provide immediate mitigation, while at the same time improving air quality (Kopp and Mauzerall, 2010). To understand the tempo-spatial distribution of BC mass and associated absorbing properties is the key to resolve its radiative forcing impact in the atmosphere.

The single particle measurements of BC have been recently conducted on the ground over East Asia (Huang et al., 2011;Wang et al., 2016b;Wu et al., 2017;Gong et al., 2016). Besides BC mass loading, the size distribution and mixing state of surface BC was also available in these studies for a range of megacities. The BC emitted from ground sources can be efficiently transported upwards through the convective mixing of the planetary boundary layer (PBL). Increasing studies pointed out that the severe haze events occurring in East

Asia were mostly associated with the unfavorable meteorology condition rather than enhanced emission (Sun et al., 2015;Zheng et al., 2015). Many modelling studies (Zheng et al., 2015;Zhao et al., 2013;Zhang et al., 2015;Liu et al., 2013b) suggested the large presence of aerosol layer in the lower atmosphere would prevent the downwelling solar radiation reaching the surface, and this surface dimming effect could depress the development of the PBL, increasing the occurrence of heavy haze events in East Asia. This unfavorable

weather condition will be further exacerbated if significant presence of BC in the PBL (Ding et al., 2016), as the strong heating effect of BC in the lower atmosphere could enhance the atmospheric stratification and further decrease the PBL height. Characterizing the distribution of BC mass and its absorption in the lower atmosphere is therefore crucial to determine its regional impact through modifying the atmospheric thermodynamics.

The measurement on the vertical distribution of BC in the polluted boundary layer over East Asia is sparse. The only information was from measurements through a mirco-aethelometer equipped in weather balloon (Ran et al., 2016), the offline microscopy analysis (Zhang et al., 2012) and recent aircraft measurement using a single particle soot photometer during a summertime (Zhao et al., 2015). There is still lack of characterization of BC throughout the atmospheric column with sufficient seasonal coverage, and the vertical

profiles of BC size distribution and mixing state which importantly determine its absorbing properties (Liu et al., 2017) have not been widely reported. The in-situ measurement of BC properties in the boundary layer over the polluted region in East Asia is thus desired to understand its feedback on meteorology pattern and the consequent influence on the accumulation of pollutants.



This study presents the results of in-situ measurements of BC from the King-air and Y12 aircraft platform (Beijing Weather Modification Office), during 10 flights in late spring season (Apr.-Jun. 2012) and 6 flights in winter (Dec. 2016). The flights were conducted over the North China Plain (NCP), one of the most polluted and populated region in East Asia. The flights took place mostly in the boundary layer over Beijing

and the surrounding area. Fig. 1 summarizes all of the flight tracks conducted in both seasons. As the BC emission inventories over the NCP (Zhang et al., 2009) show (also in Fig. S1), the high anthropogenic BC emission can be generally divided by the border along Taihang and Yanshan Mountains, beyond which the region from northwest Beijing has relatively lower emissions. As Bejing is at the northern edge of the NCP, the emissions are likely to be accumulated at the foothill of the northern mountains if the air mass comes

from the south. In the emission inventory, the residential sector mostly including the coal usage for residential heating during cold season has remarkably higher BC emission than in hot season by an order of about 10. The industrial and transportation sectors almost maintained throughout the year. This means the source difference of BC observed during cold and hot seasons will mainly stem from the different contribution of residential heating. For both seasons, the polluted air mass will tend to come from the south,

with particularly high emission from the southwest, but the air mass from the north and northwest will be typically clean. There are significant local BC emissions from all sectors surrounding Beijing throughout the year.

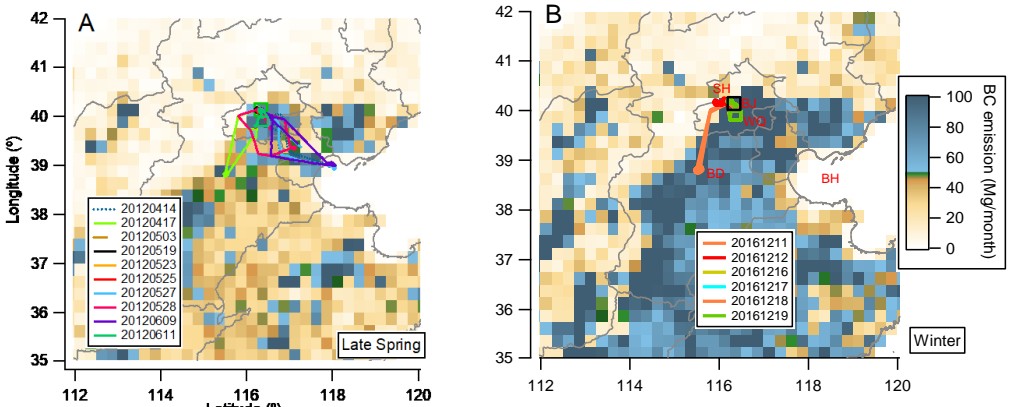

Fig. 1. The flight tracks for late spring and winter, as denoted by flight date. The open square marks the
location of central Beijing. The abbreviations on the right panel show the location of major cities, BJ (Beijing), SH (Shahe), BD (Baoding), WQ (Wuqing) and the ocean, BH (Bohai). The map is imaged by 2010 BC total emission inventory (the sector-segregated emissions are shown in Fig. S1).



## 2 Aircraft platform, instrumentation and data analysis

The aircraft platforms employed in this study are shown in Fig. S2. The data in 2012 was collected by Yun-12 aircraft, which operated at an average speed of 200 km/hr. The 2016 flight was performed on King-Air350 board which has a higher operation speed and improved ceiling. The sampling inlet system is

the Model 1200 passive Isokinetic Aerosol Sampling Inlet (BMI, Brechtel Manufacturing Inc) shown in Fig. S3, which was designed to deliver 150 lpm of sample flow with 100 m/s air speed and is able to transmit particle diameters 0.01-6 μm with > 95% collection efficiency. The operation of most flights attempted to avoid the clouds, and even if in cloud the results presented here have screened out the in-cloud condition according to the measurements of relative humidity and cloud liquid water content. The flights were mostly

operated at altitude up to 3600m focusing on the pollutants in boundary layer over Beijing. The flight track and time schedule were almost consistent for all flights (supplement Fig. S7): The aircraft departured from Shahe in the morning (a rural area on the northwest about 20km away from central Beijing), a full profiling was conducted and then flied over the Beijing city or the surrounding area performing a few constant-level runs at different altitudes and followed by another full profiling over Shahe before landing in the late

afternoon. The consistent time schedules (departure ~9:00-10:00, return ~11:00-14:00 LST during late spring; departure ~12:00-13:00, return ~15:00-16:00 LST during winter) avoid the possible diurnal variation of the PBL development among different flights, and all the profiles were conducted under the daytime well-developed boundary layer. The flight track was assigned with different geographical locations over the major cities with a 5km×5km covering area from the city center (shown in Fig. 1B): Shahe (SH), Beijing

(BJ), Baoding (BD), Wuqing (WQ), the eastern ocean Bohai (BH), and the rest is where the aircraft was in transit between the locations. Most of the full profiles through the surface and the PBL were performed over SH city, as the aircraft was unable to reach lower level below 500m in other cities due to the regulation of flight control. The height of PBL (PBLH) in this study is therefore mainly determined over Shahe where the full profiling was performed.

The meteorological parameters including ambient pressure, temperature, relative humidity and wind speed/direction were characterized by AIMMS-20 (Aircraft Integrated Meteorological Measurement System, Aventech Research Inc), which was calibrated on an annual basis. In addition, the stratified wind speed and direction up to 2500m altitude was measured by a ground based boundary wind profile radar (Airda-3000, Airda Co., China) located at Shahe city. Given the aircraft profiles were also conducted over the similar

location at Shahe, the wind radar data showed high consistency with the in-situ measurement (supplement Fig. S4). As the AIMMS data was only available in wintertime, the wind profile from radar measurement is used for both seasons for consistency. At Shahe, the surface particular matter smaller than 2.5μm (PM2.5) was measured by an R&P model 1400a Tapered Element Oscillating Microbalance (TEOM, Thermo Scientific Co., USA) covering the flight periods for both seasons.





The physical properties of individual refractory BC particles were characterized using a single particle soot photometer (SP2) manufactured by DMT Inc (Boulder, CO, USA). The instrument operation and data interpretation procedures for the aircraft SP2 have been described elsewhere (Schwarz et al., 2006;McMeeking et al., 2010). The SP2 incandescence signal was calibrated for BC mass using Aquadag® black carbon particle standards (Aqueous Deflocculated Acheson Graphite, manufactured by Acheson Inc., USA) and corrected for ambient BC with a factor of 0.75 (Laborde et al., 2012). The time required to evaporate the coatings on the BC showed distinct bimodal mode, and the mixing state of BC can be characterized as a number faction of BC particles with thicker coatings ($F_{coating}$) with longer evaporation time of coatings in the SP2 laser beam (Moteki et al., 2007;Liu et al., 2010). In this study, the threshold coating evaporation time to determine $F_{coating}$ was ~1μs (Fig. S5A), and the $F_{coating}$ is calculated within the most populated BC core size range at 120-180nm, to avoid the possible bias introduced from the varying core size distribution on the $F_{coating}$ calculation. In 2012, the mixing state of BC was also characterized by the ratio of particle optical size and BC core size (Liu et al., 2014;Taylor et al., 2015). Both $F_{coating}$ and optical sizing methods are well correlated (supplement Fig. S5B). As the optical sizing function of the SP2 in 2016 was not available, the $F_{coating}$ method is used for both seasons for consistency. The BC core mass median diameter (MMD), defined as a diameter below and above which the BC mass loading is equal, is used to illustrate the variation of BC core size and reflect the pattern of BC core size distribution (Liu et al., 2014).

## 3 The meteorology and the determination of layers

A typical example of vertical profiles over Shahe (flight 20120528) for the BC properties and meteorological parameters is shown in Fig. 2 (the rest flights are shown in Fig. S7). The profiles are assigned with different aircraft locations over the major cities (as Fig. 1) and are analyzed separately. For most of the flights, the BC mass showed a well mixing at lower level, and was subjected to scavenging at certain altitude ($z$). To investigate the contributions of surface emission and regional transport on the measured profiles, the in-situ or remotely measured local meteorology is firstly analyzed and then the regional transport is explored by synoptic analysis.

The surface layer is defined as below 100m in this study where the pollutants were largely influenced by the surface emissions and roughness (Antonia and Luxton, 1971). The height of planetary boundary layer (PBLH) is the layer where pollutants are efficiently uplifted from the surface through convective mixing. The PBLH in this study is firstly determined by the vertical gradient of the in-situ measured virtual potential temperature ($\theta_v$). As Fig. 2A and H shows, the $\theta_v$ had little vertical gradient at lower level and at certain altitude exhibited a positive gradient, and then reduced at higher altitude. In this study, the altitude ($z$) at which the vertical gradient d$\theta_v$/d$z$ reached 5K/km was defined as the PBLH, below which the d$\theta_v$/d$z$ was less than 5K/km denoting a thermal-dynamically well mixed layer. As Fig. 2D-F shows, the PBL was characterized as less





variation of BC mass, RH and the mixing state of BC ($F_{coating}$). To further evaluate this definition, the vertical wind profile measured at Shahe combined with the measured $\theta_v$ is used to calculate the bulk Richardson number ($Ri_b$) to determine the PBLH (Hong et al., 2006), as expressed by Equation (1):

$$Ri_b(z) = \frac{\left(\frac{g}{\theta_{v,s}}\right)\Delta\theta_{v,z}\Delta z}{\Delta u^2 + \Delta v^2 + bu_*^2} \quad (1),$$

The $Ri_b$ is calculated at every 50m of altitude ($z$), the $\theta_v$ and vertical profiles of horizontal wind (eastward $u$ and northward component $v$) are from the wind radar measurement. The data is averaged at 50m altitude bin, and the $\Delta\theta_v$, $\Delta u$ and $\Delta v$ is calculated to be relative to the average value in the surface layer ($z$=100m); $g$ is the gravity acceleration constant; $u*$ is the surface friction velocity, which is omitted in this study given the calculation starts from $z$=100m and this value is small compared with the wind shear term.

The $Ri_b$ essentially includes the influence of both surface thermal intrusion and vertical wind shear, which is dimensionless and a lower $Ri_b$ means stronger air vertical turbulence. The $Ri_b$ usually increases with altitude, and the altitude at which the $Ri_b$ reaches a critical value of $Ri_b$ ($Ri_{b,c}$) defines the top of PBL, as widely used in chemical transport model to work out the PBLH (Skamarock et al., 2005;Hong and Lim, 2006), however the $Ri_{b,c}$ may depend on the environment and the exact air conditions (Zhang et al., 2014;Guo et al., 2016).

Fig. 2H gives an example of the calculation of $Ri_b$ (the open round marker shows), showing that the $Ri_b$ reaches $Ri_{b,c}$=0.25 at a similar altitude where d$\theta_v$/d$z$ reaches 5K/km. This suggests the consistence between the two methods in determining the PBLH, however the unstable wind profile data at certain altitude, such as due to the possible topographical influence, will cause some erroneously high $Ri_b$ (Fig. 2H), leading to difficult automatic calculation of PBLH based on measurements. The first few high $Ri_b$ is thus removed at

lower level and then the PBLH is calculated as the level where the $Ri_{b,c}$ firstly reaches. The determinations of PBLH for the other flights are shown in Supplement Fig. S7. Fig. 3 gives the comparison between the two methods for all flights and shows ±10% difference between both methods. In late spring, the choice of $Ri_{b,c}$=0.25 gave slightly lower PBLH but a $Ri_{b,c}$=0.30 best agreed with the d$\theta_v$/d$z$ method.



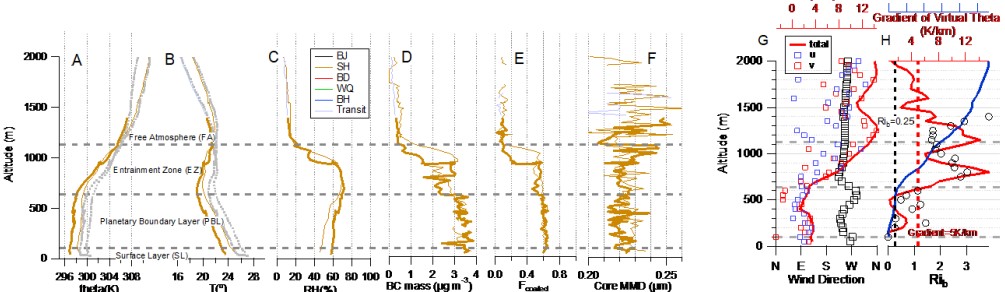

Fig. 2. A-F) Vertical profiles of potential temperature (θ), ambient temperature (T), relative humidity (RH), BC mass, $F_{coating}$ and BC core MMD for flight 20120528, with the grey dashed lines showing the virtual potential temperature ($θ_v$) and virtual temperature ($T_v$); G) horizontal wind speed and direction; H) The criteria to determine the PBLH using vertical gradient of $θ_v$ and critical bulk Richardson number. The black and red dashed line indicate the $Ri_b=0.25$ and $dθ_v/dz=5k/km$ respectively. The horizontal dashed grey lines show the determined heights of each layer.

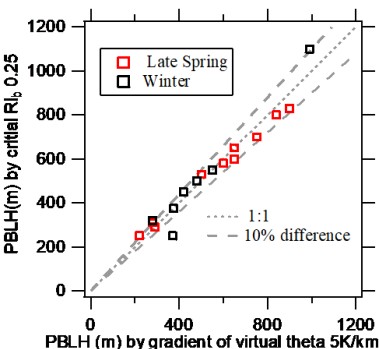

Fig. 3. The PBLH determined by vertical gradient of virtual potential temperature ($θ_v$) method as a function of that determined by critical bulk Richardson number.

On top of the PBL, there was usually a temperature subsidence inversion (Fig. 2B) which stabilized the mixing layer below. The degree of this temperature inversion varied among flights and depended on the local meteorology on that day. This layer up to the bottom of free atmosphere (FA) is deemed to be the entrainment zone (EZ) (Sullivan et al., 1998). The EZ generally featured with a decrease of RH (Fig. 2C) and aerosol concentration, but increased $θ_v$ and usually increased wind shear (Fig. 2G). In the EZ, the aerosols are subject to efficient scavenging or horizontal transport. The lower level clouds such as cumulus or stratus are usually initialized from this layer (Miles et al., 2000;Siems et al., 1990), and the aerosols may have been scavenged by cloud particles. The top of EZ is determined as the level where BC mass loading decreased to 5[th] percentile in that profile, and for most of the flights the top of EZ coincided with the diminishing of the



temperature inversion on top of PBL. Above EZ it is the FA layer until the flight operation level up to 2.5-3km, where the BC mass was generally low and had no much variation.

In hot season, the return legs of a few flights in the early afternoon showed dynamic meteorological conditions compared to the departure over Shahe (Fig. S6A). During return, the surface temperature increased and the profile patterns showed strong instability of the atmospheric column: the $\theta$ decreased with altitude, the temperature deceased with altitude but without any inversion, RH and BC mass dramatically dropped, and wind speed increased. The $d\theta_v/dz$ showed consistently <5K/km up to the upper height limit of wind radar 2km; the $Ri_b$ was scattered and as low as 0.3 at 2km (Fig. S6B). The PBLH is therefore unable to be identified within $z$=2km by using neither of the methods, which in turn demonstrates that the atmosphere was highly unstable for this case. For the following discussions, such weather condition is termed as turbulent (T), and all information observed in profile is deemed to be within the highly convective PBL. The exact PBLH may be found at $z$ >2km, but the information of PBLH for this highly turbulent air condition will not be discussed in this study. Both the aerosols and moisture were significantly dispersed both vertically and horizontally, thus showing low throughout the column. It is noted that the development of PBLH in late spring was not in a consistent diurnal pattern, e.g. not all of the return legs had a more developed PBL than departure, which means the diurnal variation of surface temperature is not the only factor in determining the PBLH but also the synoptic weather system.

Fig. 4 shows a few examples of synoptic conditions for the periods when PBL was influenced by different local wind directions (the other flights are shown in Fig. S7). The wind field at 925mbar (which corresponds with the average PBLH) showed that when southerly air mass, the synoptic wind direction could be modulated by the mountain lining along SW-NE when the air mass reached the foothill (Fig. 4A1-2, B1-2), resulting in dynamic local wind directions; whereas when northerly air mass, this topographical influence on wind was less (Fig.5 A3 and B3). For both seasons, when the center of high pressure system was located off the eastern coast, the predominant southwest flow would bring the most polluted air mass along the Taihang Mountains (Fig. 4A1 and B1). The atmosphere was more stabilized when Beijing was located in a saddle between the centers of two high pressure systems (Fig. 4A2), resulting in much lower wind speeds. The wind field at 925mbar was mostly consistent with the local wind profile measured over Shahe (Fig. S7).

In addition, the HYSPLIT backward trajectory model (Draxler and Hess, 1998) was run over the location of Shahe at 500m a.s.l. for the past 48 hours using the 1°×1° horizontal and vertical wind fields provided by the GDAS1 reanalysis meteorology. Note that the real time backtrajectory analysis initialized along the flight track was also performed for each flight, however there was no apparent variation of backtrajectory pattern within the PBL for all flights, thus only backtrajectory initialized at 500m is shown here. As Fig. S7 shows, the measured wind directions in the PBL agreed well with the directions of recent 12h backtrajectories for most flights, apart from a few flights showing some discrepancy between both methods when low wind speed, which may result from the topographical influence on the local wind (such as flights 20161217 and



20161218). Some backtrajectories of more than 12h backwards shifted to different directions compare to recent 12h (such as flights 20120525, 20120527 and 20120609). Given the intensive local emissions (Fig. 1) and significantly overlapped emissions from different sectors around Beijing (Fig. S1), the resolution of the wind field reanalysis data as the input of backtrajectory analysis (1°×1°) may not be able to ideally allocate

5    the detailed emission sources, this study thus combines the backtrajectory within 12h and the measured local wind profile to identify the broad direction of air mass (which mostly influenced the pollutants in the PBL).

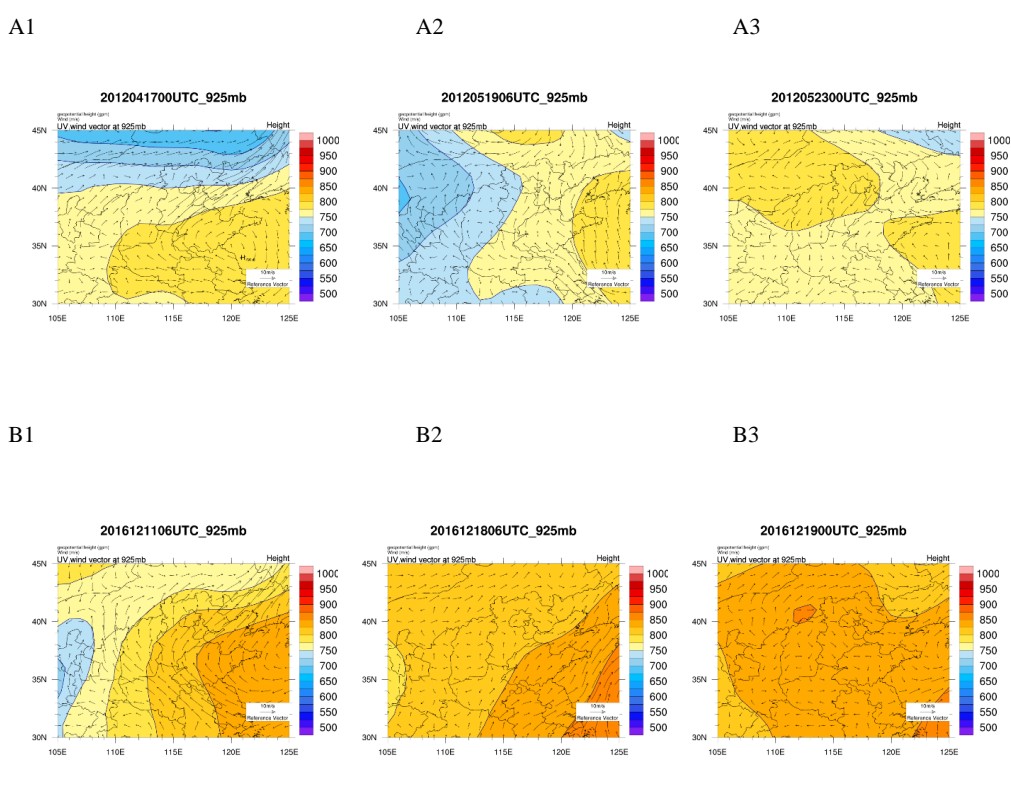

Fig. 4. Synoptic wind field at geopotential pressure 925mb. A1-A3) late spring flilghts when Beijing area was influenced by SW, S/SE and N/NW wind respectively; B1-B3) winter flights, Beijing was influenced by SW, low SW/S wind speed and NW wind respectively.

15    4 Results and discussions

4.1 Overview of the vertical profiles and layer-segregated results for both seasons

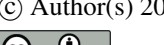


Fig. 5 summarizes the vertical profiles of temperature and BC properties for all flights in both seasons. The black lines with open square show the project average excluding the turbulent weather conditions (marked in red). It shows remarkable differences for the vertical structures of BC properties between both seasons. The hot season showed a generally more developed PBLH and less vertical gradient of BC mass in the PBL

compared to winter, whereas in winter the BC mass was accumulated at lower levels. The turbulent weather condition in hot season showed missing temperature subsidence inversion, dramatically reduced BC mass and $F_{coating}$; under the dynamic air condition in winter (flight 20161211), both locally and regionally transported BC mass was uplifted or convectively mixed through a much more developed PBL up to 1km, and the $F_{coating}$ of BC had not been reduced even in the morning of next day (section 4.2). The BC core size

in late spring showed to be systematically lower than in winter. The following sections will in detail discuss the BC properties in the PBL (section 4.2) and explore the reasons causing the difference of BC properties between seasons (section 4.3).

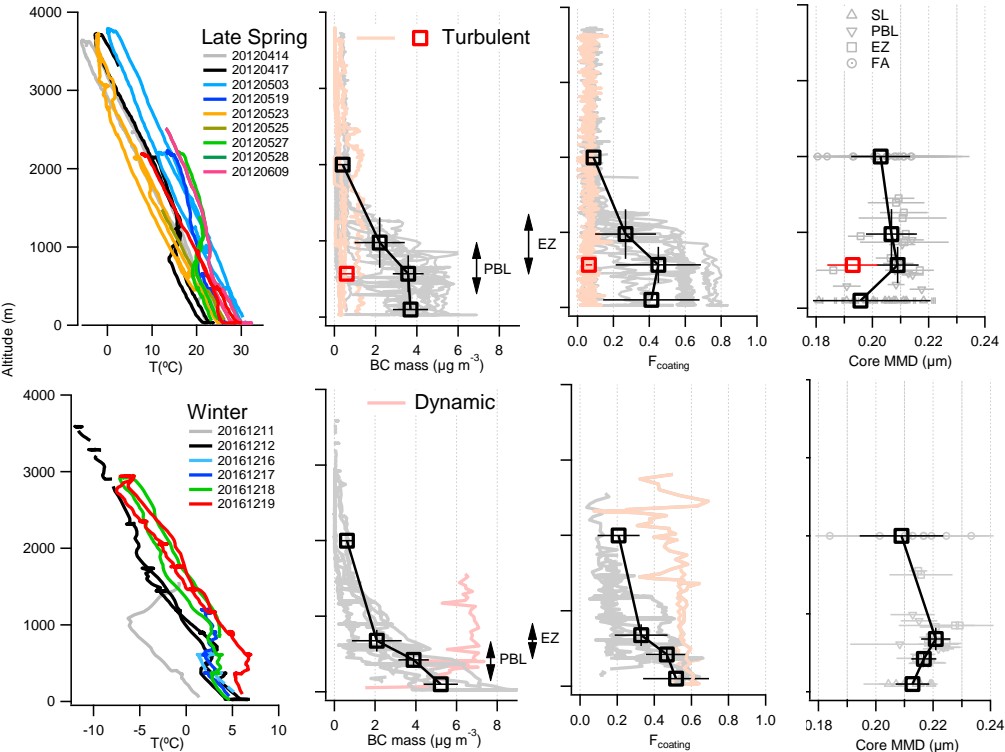

Fig. 5. The vertical profiles of temperature (coloured by flight index) and BC properties (separated by
turbulent or dynamic weather conditions) for all flights. The BC core MMD is shown as the mean ± standard deviation at different layers for all flights. The thicker black lines with open squares show the project average of BC properties in the surface layer (SL), planetary boundary layer (PBL), entrainment zone (EZ) and free





troposphere (FA) over all flights, the y-axis of which shows the project average height of PBL and EZ with vertical error bars showing ± standard deviation (here the height of SL and FA are set at fixed 100m and 2000m respectively). The double-end arrows show the variations of the PBL and EZ heights among flights. The red lines and markers show the results under all the highly turbulent weather conditions observed in late

spring (when the PBLH is not determined) and the winter flight 20161211 with significantly high PBLH, and these are excluded from the calculation of the project average.

The averaged BC properties in the defined layers and the corresponding PM2.5 on the ground covering the flight periods are shown in Fig. S8. At the same PBLH, the late spring showed a thicker EZ than winter (Fig. 6A), because the higher surface temperature in hot season caused an enhanced thermal buoyance of the

uplifted air mass, with strongly developed PBL penetrating to the EZ. The BC mass in the PBL ($BC_{PBL}$) generally co-varied with the surface PM2.5 for both seasons (Fig. 6B). As Fig. S7 shows, the flights in late spring covered a wide range of pollutant levels from very clean ($BC_{PBL}<1$ $\mu gm^{-3}$) to polluted days ($BC_{PBL}>4$ $\mu gm^{-3}$). The flights were not conducted in the very clean days in winter (14 to 16 Dec.). The winter flights were conducted at a higher pollution level than in late spring with many days the surface PM2.5 exceeding

160 $\mu gm^{-3}$ in winter. At the same level of PM2.5, the $BC_{PBL}$ in late spring was higher than in winter by a factor of about 1.5 (Fig. 6B). This may be due to the higher contribution of residential sector which emitted higher fractions of organic matter causing lower BC fractions in cold season (Yang et al., 2005;Cao et al., 2005). The stronger convective mixing of surface sources in hot season led to a higher pollutant concentration in the PBL than at surface, which also corresponded with a slightly higher BC mass in the EZ

($BC_{EZ}$) in late spring than in winter at the same level of $BC_{PBL}$ (Fig. 6C). Note the flight 20161211 and the beginning of flight 20161212 in winter were conducted under highly polluted days with much stronger developed PBLH, which are marked separately.

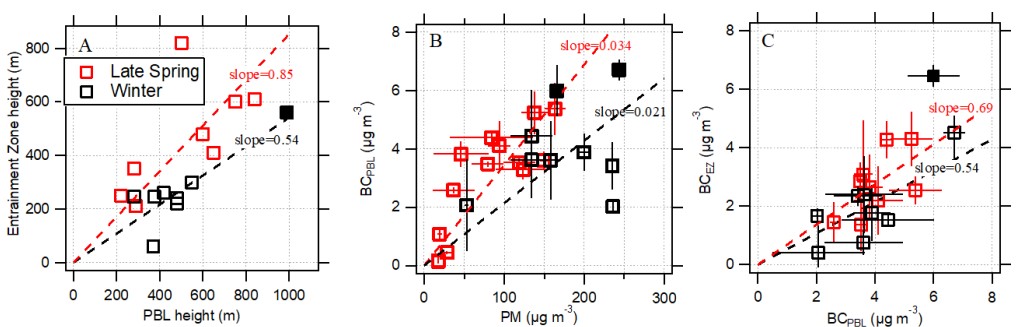

Fig. 6. A) The EZ height vs PBL height for all flights (note that the PBLH for highly turbulent weather
condition during late spring is not determined); B) The BC mass loading in the PBL as a function of surface PM2.5 averaged over the flight period; C) the BC mass in the EZ vs in the PBL. The flights conducted at the highly dynamic air condition in winter were marked as the solid black square.

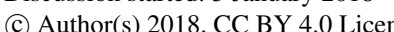



## 4.2 The BC mass loading in the PBL

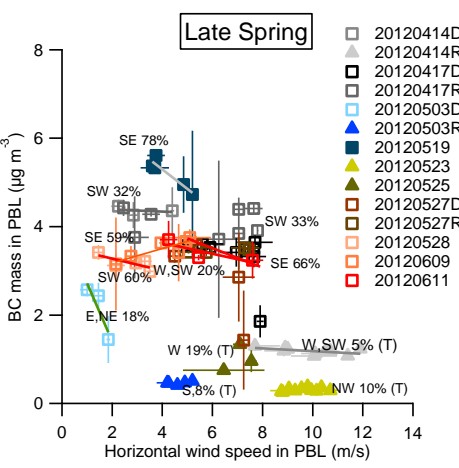
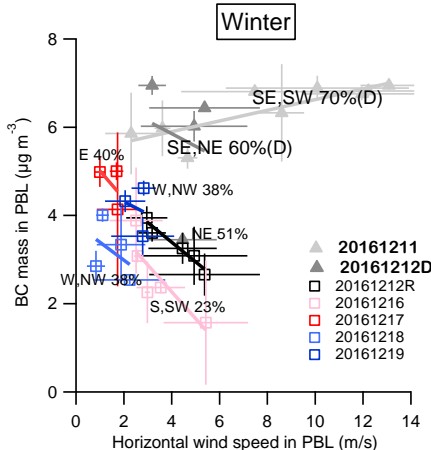

Fig. 7. The BC mass as a function of horizontal wind speed in the PBL for all flights in both seasons. The legends on each flight show the air mass direction (from measured wind profile and backtrajectory analysis) and RH value in the PBL. The highly turbulent weather condition is marked as solid triangle for late spring and denoted as (T); in winter the dynamic air condition is marked as (D). The wind speed and BC mass loadings are the mean value averaged in 100m attitude bin, with error bars showing ±standard deviation. The lines show the least-square linear regression for each flight.

The BC mass loading related to wind speed and direction in the PBL is shown in Fig. 7. Combing the local wind profile and synoptic analysis (Fig. S7), in late spring the air masses were mostly southerly whereas in winter the air masses mostly came from the north. The $BC_{PBL}$ was not only influenced by emissions, i.e. the air mass from the more polluted south in particular the southwest direction would bring more BC mass, but also importantly determined by meteorology.

In late spring, most of the lower wind speed (<8m/s) was associated with the air masses from SW or SE direction, and under these conditions, $BC_{PBL}$ always exceeded 3µg m$^{-3}$, apart from one flight 20120503 encountering a lower BC mass loading with NE/E air mass direction. For all flights in late spring, the dilution effect of wind was not obvious, shown as the low gradient of linear regression fitting (between $BC_{PBL}$ and horizontal wind speed) in Fig. 7. With increasing wind speed at higher level in the PBL, the $BC_{PBL}$ showed little vertical gradient (Fig. 7 and Fig. S7). This suggests the strongly mixed and well developed PBL in hot season where the pollutants were efficiently uplifted through convective mixing. In addition, the dominant SW/SE air mass direction during polluted days may have also advected the pollutants from the polluted south (Fig. 1), which may offset the dilution effect of higher wind speed. The air mass from the south also contained more moisture which contributed the higher RH at polluted days. The highly



turbulent weather conditions (marked as T) as mostly observed when return flight in late spring (Fig. 4) featured with higher wind speed (4-12m/s) and dynamic air mass direction. The wind under such weather conditions had significant dilution effect on both pollutants and moisture throughout the atmospheric column, leading to a much reduced BC mass loadings <2µg m$^{-3}$ and RH <20%. Note that under highly turbulent air

condition even the southerly wind (return flight on 20120414 and 20120503) had not enhanced the $BC_{PBL}$ which means the thermodynamic influence may overcome the contribution from regional transport.

During the winter flight 20161211, the high pressure system at 925mbar centered off the eastern coast (Fig. S7) led to strong SW/SE wind in the PBL, which efficiently advected the highly polluted air mass from the south. The transport of warmer air from the south quickly imposed on the cold air mass from previous days,

resulting in a much enhanced RH ~70%. The extremely high wind speed (6-14m/s) under such dynamic air condition had not diluted the pollutants but even further promoted the well mixed BC mass layer up to 1.1km and uplifted the BC in the EZ up to 1.6km. In the next day (flight 20161212), though the synoptic pattern of wind shifted as from the north, the high moisture (RH=60%) maintained on the departure flight and the BC mass loading was still high in the PBL (>5.5 µg m$^{-3}$). The $F_{coating}$ of BC in the PBL and EZ maintained as

high as ~0.6 consistent with the previous day (Fig. 5) but only altered in the afternoon. Both flights demonstrated an entire process for the accumulation and removal of pollutants in the boundary layer. The meteorology started to change since the return of flight 20161212 with wind direction shifting northerly and with much reduced wind speed (<6m/s). This northerly or northwesterly synoptic air mass direction maintained throughout the rest of the flights according to the backtrajectories (Fig. S7), though the local

wind profiles showed some divergence on wind direction in the PBL. Due to the clean N/NW air mass, the pollutants were deemed to be mainly contributed by local sources during this period, which also corresponded with the reduced $BC_{PBL}$ compared to the first two flights, given there was no significant contribution of regional transport from the south. For all flights during 16-19 Dec., the lower wind speed towards the surface corresponded with a higher BC mass loading accumulated at lower level, and the $BC_{PBL}$

showed strong vertical gradient and anti-correlation with wind speed, suggesting the efficient dilution effect of high speed wind.





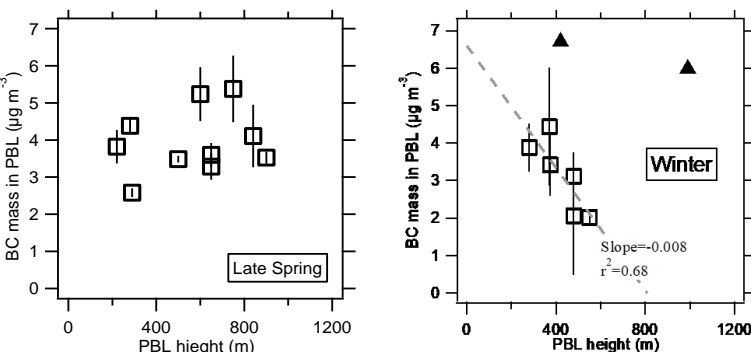

Fig. 8. BC mass in the PBL as a function of PBL height (PBLH) from all flights. The linear fitting in winter excludes the dynamic air condition which is marked as solid triangle. Note that for late spring, the PBLH was not determined for highly turbulent air condition.

The influence of wind pattern on $BC_{PBL}$ was in line with its correlation with the PBL height (PBLH), as shown in Fig. 8. In late spring, because of the well-mixed PBL and regional transport from the south, the BC had been well mixed in the PBL and there was no apparent correlation with the PBLH. In winter, the $BC_{PBL}$ was obviously anti-correlated with the PBLH at a decreasing rate of ~8µg m$^{-3}$ per 1km increase of PBLH, apart from the heavily polluted days in dynamic air condition with significant regional contribution (solid

marker in Fig. 8B).

Combing the results from both seasons, it can be concluded that the BC mass in the PBL is a combination of the accumulation of local emissions and contribution from regional transport. The frequent occurrence of SW/SE wind in late spring retained $BC_{PBL}$ at 3-5µg m$^{-3}$, because the dilution effect of high speed SW/SE wind may have been compensated by the efficient transport of regional pollutants from the polluted south.

The pollutants in late spring were thus likely to be a combination of fresher local emissions and transported aged particles. The $BC_{PBL}$ had not been significantly modulated by the PBLH which may be partly due to this regional influence. The strong transport of regional pollutants from the south was also observed in winter, when the BC mass had been remarkably elevated throughout the PBL even under high wind speed >6m/s.

The cleaner air mass from NW/N/W had apparent dilution effects, which may be the main driver that in

winter the $BC_{PBL}$ was strongly anti-correlated with the wind speed and the high BC mass loading was accumulated near the surface. This in turn suggests for most of the time the local emission played an important role during winter. The $BC_{PBL}$ was thus importantly modulated by the PBLH given there was no significant regional input. The remarkable dilution effect when highly turbulent air condition was frequently observed in early afternoon during late spring, with the horizontal wind speed >6m/s regardless of the air

mass direction (even for southerly air mass direction of flight 20120414 and 20120503), when the PBL was fast developed and reduced the BC mass <2µg m$^{-3}$. This means within highly turbulent and convergent





atmospheric column, the dilution of wind overwhelmingly took over the accumulation of local emissions or regional transport.

4.3 The BC size and mixing state at different layers

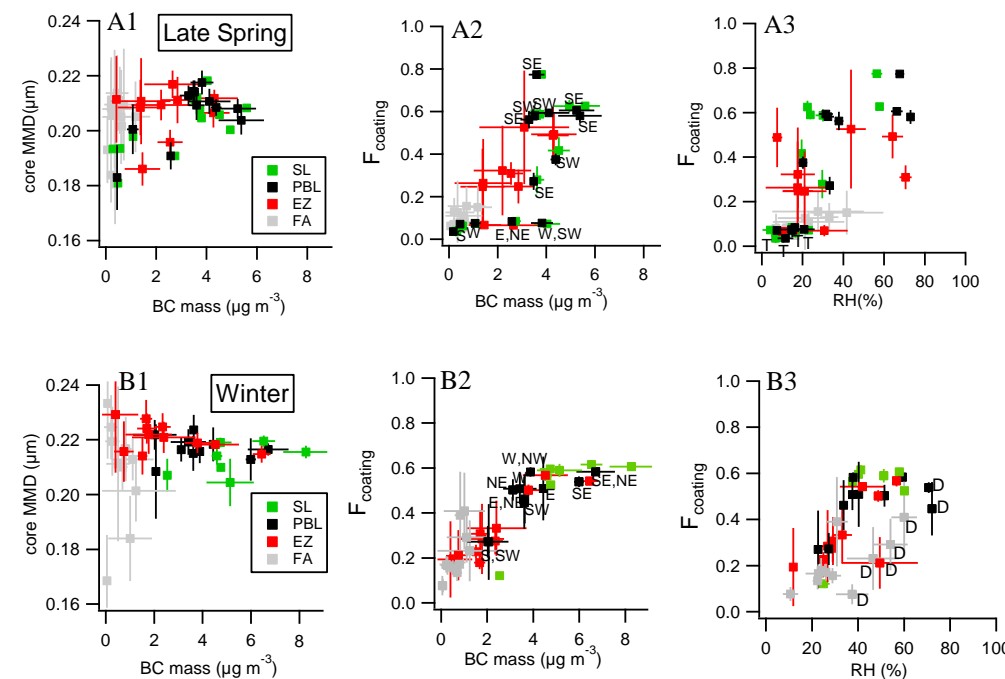

Fig. 9. The relationship among BC properties and RH at different layers for all flights. Each maker with error bars denotes the mean ± standard deviation at different layers of each flight. The air mass direction in the PBL is marked in panel A2 and B2. The highly turbulent or dynamic air condition is marked as T or D in

10 panel A3 and B3. Note this figure also includes the flight runs over all major cities (Shahe, Beijing, Baoding and Wuqing).

The correlations among BC properties and RH at different layers are shown in Fig. 9. The project average over all flights is shown in Fig. 5 and Table 1. As Fig. 9A1 and B1 shows, the BC core size in SL/PBL/EZ in late spring had wider diversity than in winter. For the BC observed in the surface layer (SL) and PBL, the

15 majority of the BC core size in terms of mass median diameter (MMD) is 0.21-0.23µm in winter; besides this major MMD mode, in late spring, smaller mode of MMD was observed at 0.18-0.20µm at low BC mass (<3µg m⁻³) under highly turbulent weather. The BC core size was previously observed to be source-dependent, such as the BC from coal burning in urban Beijing (Wang et al., 2016b) or wood burning in urban London (Liu et al., 2014) had a larger core size than from traffic source. For the project average (Fig.



5 and Table 1), the BC core size was systematically lower in late spring, which may be due to a lower contribution from residential sector in hot season. The BC core size slightly decreased when BC mass >5µg m$^{-3}$ when SE air mass direction (Fig, 10A1), which may result from a more contribution of traffic source when SE air mass (Fig. S1). The less variation of BC core size in the PBL during winter may be due to the overwhelmingly dominant contribution from residential sector. The BC$_{PBL}$ consistently had larger core size than that in the SL for both seasons, which may result from coagulation process or because in the PBL, the BC had experienced a more homogenous mixture of primary sources from different emission sectors.

The BC core size in the entrainment zone (EZ) also had different seasonal behavior. In summer, the BC core size in the EZ was mostly smaller than in the PBL, whereas in winter the core size in the EZ was often higher than that in the PBL. This means the scavenging of BC in the EZ had not significantly modified the core size in winter, but in late spring the BC containing larger core was favorably removed (Moteki et al., 2012). The BC in the free atmosphere (FA) had been significantly removed and may contain BC from long range transport. A lower BC core size in the FA was also observed in previous studies (McMeeking et al., 2010) which indicates that the background BC after significant atmospheric scavenging would mainly contain smaller particles.

As Fig. 9A2 and B2 show, at polluted days (BC$_{PBL}$>3µg m$^{-3}$), the BC$_{PBL}$ in winter had a uniform $F_{coating}$ of about ~0.5-0.6, whereas in summer besides a mode containing a few flights with a $F_{coating}$ ~0.6, a wider diversity of coating content was also observed. The mode with a consistent $F_{coating}$ ~0.6 was in line with the consistent core MMD ~0.21µm. This mode of BC may result from a homogenous mixing of sources when high level of pollution. In late spring, the less coated BC with $F_{coating}$ spanning from 0.05-0.4 was mostly associated with RH <30% (Fig. 9A3); Fig. 9 A2 shows these BC particles were either from the background after significant removal of BC by highly turbulent air (when BC mass <2µg m$^{-3}$) or from primary emission (BC mass >2µg m$^{-3}$), however the high BC mass loading with low $F_{coating}$ was not observed in winter. This may indicate a large content of BC coatings from residential sector in winter but in summer the fresh urban or industrial plume may occasionally dominate. The $F_{coating}$ in the EZ was consistently lower than in the PBL for both seasons. In the FA, the $F_{coating}$ was <0.2 in late spring but a higher $F_{coating}$ in the FA during winter was from the dynamic air when significant regional transport.

For both seasons, the $F_{coating}$ was generally positively correlated with RH in the PBL (Fig. 9A3 and B3). For both seasons, a rapid increase of $F_{coating}$ took place when RH increased up to 40%; as long as RH >40%, the $F_{coating}$ reached ~0.6. This may result from the enhanced heterogeneous or aqueous reaction to form the secondary aerosol species at high RH (Sun et al., 2006), or the regional transport advected both primarily thickly coated BC and high moisture. For the dynamic air condition in winter (marked as T in Fig. 9B3), a lower $F_{coating}$ exhibited at the same level of RH compared to normal days, which may suggest the fast transport of BC with moisture air that the BC had not been sufficiently coated during transport due to the short time scale of ageing.



Combing the results in Fig. 9, it showed that the enhanced BC mixing state mostly took place when high RH and high BC mass loading. This suggests either the importance of moisture on promoting the formation of secondary coatings on BC, or the southerly air mass containing more moisture may have transported a large amount of BC such as from residential sector which contained considerable amount of coatings. In reality,

both factors will contribute but only chemical analysis on the coatings (Wang et al., 2016a) may rule out whether primary or secondary process may dominate. By comparing the polluted and clean days, it can be indicated that the removal of more coated BC and the moisture may take place at a similar time scale, leading to most of the clean and dry air containing less coated BC.

**4.4 The removal of BC in the entrainment zone**

Significant removal of aerosols takes place in the entrainment zone (EZ). The aerosols in the EZ are subject to removal by increased dilution, as in this study most of the EZ had apparent wind shear with wind speed 5-10m/s higher than in the PBL below. The EZ is where the low-level cloud is initialized (Miles et al., 2000), thus some of the aerosols may have been scavenged by incorporating into the cloud particles or by impaction

removal (Han et al., 1998;Albrecht, 1989). The aerosols may be continuously uplifted by the cloud updraft or moisture convection, and the processed aerosols after scavenging will eventually represent the background in the free atmosphere. Assuming the BC in the EZ was mainly contributed by the BC in the PBL below, as well as a similar influence of regional transport in the PBL or EZ, here the difference of BC mass between the PBL and EZ could be deemed to be a general indicator of the removal efficiency of BC in the EZ.

The removal efficiency of BC in the EZ, defined as $(BC_{PBL}-BC_{EZ})/BC_{PBL}$ exhibited a wide range up to 80% for both seasons. As Fig. 10 shows, the $F_{coating}$ of BC in the EZ was lower than in the PBL for both seasons. The difference of $F_{coating}$ between the EZ and PBL was positively correlated with the removal efficiency of BC in the EZ (apart from two flights in late spring with very low $F_{coating}$ in the PBL). This suggests a strong influence of BC mixing state on its scavenging efficiency, with the highly coated BC more preferentially

removed through the EZ. This may be due to the enhanced hygroscopicity of BC when coated with secondary species (Liu et al., 2013a). The BC core size between PBL and EZ however showed seasonal difference. In winter, there was no systematic difference of BC core MMD, and in many flights the BC core size was larger in the EZ than in the PBL; whereas in late spring, the BC core size in the EZ was significantly smaller than in the PBL, and the core size difference showed a positive correlation with the BC

removal efficiency in the EZ. Overall, in late spring, the BC coating and core size significantly decreased in the EZ, suggesting a much reduced particle size of BC when being uplifted into the EZ, and the larger particle had been scavenged. However in winter, the coating change was at a lower rate (relative to the removal efficiency) and no apparent change for the BC core size, indicating a moderate particle size change in the EZ.



These may be because of the relatively uniform and larger BC core size in winter, and most of the BC-containing particles may have reached the activation critical size allowing efficient scavenging take place; whereas in late spring, given the more diversity of BC core size and mixing state, only the larger BC could be efficiently scavenged, leading to a more dramatic change in BC size after scavenging. The cloud

phases, such as the presence of ice clouds in winter (as the freezing level was mostly within the EZ during winter as shown in Fig. 5) may also play different roles on scavenging BC, compared to the dominated warm clouds in late spring.

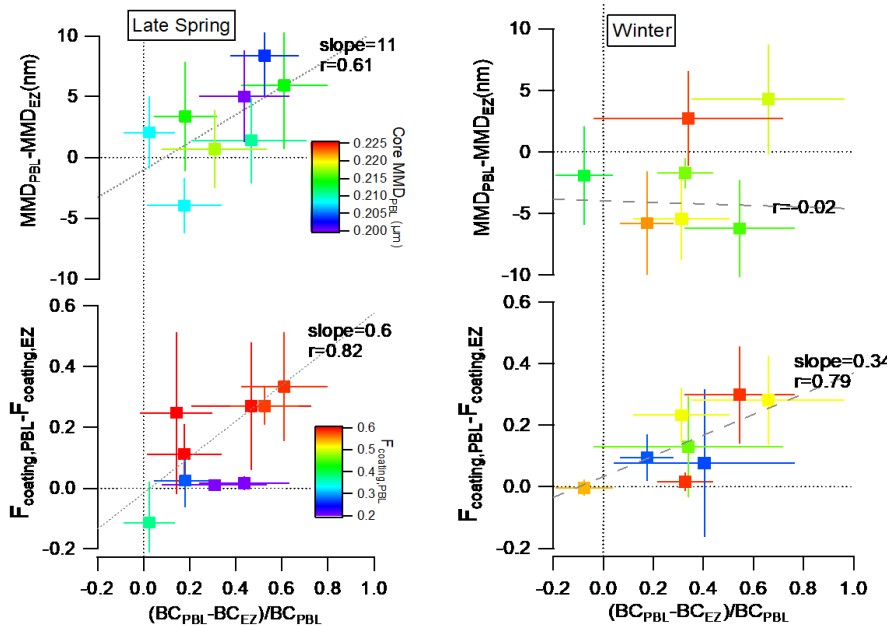

Fig. 10. The change of BC core size and $F_{coating}$ between in the PBL and EZ as a function of BC removal

efficiency in the EZ. The dash lines show the least-square linear regression. Note the fitting in late spring excludes the two points with very low $F_{coating}$ in the PBL.

## 5 Conclusion

The black carbon in the boundary layer over the North China Plain (NCP, typically around Beijing city) was

characterized by in-situ aircraft measurement in late spring and winter. The meteorology was found to modulate the primary emission and regional transport, leading to different vertical profiles in both seasons. The planetary boundary layer (PBL) in hot reason was more developed than cold season due to the higher surface temperature, leading to a deeper PBL, entrainment zone (EZ) and a better mixing of BC mass in the





PBL ($BC_{PBL}$). The $BC_{PBL}$ had less vertical gradient in late spring but decreased at higher level in winter. The more homogenous mixing of $BC_{PBL}$ in hot season may be because of the stronger surface thermal intrusion but may be also attributed to the predominant southerly wind which advected the polluted air mass containing vast emissions of residential and industrial sources; whereas in winter the cleaner northerly air

mass may play an important role on diluting the pollutants, thus most pollution was accumulated towards the surface due to reduced wind speed. There was occasional highly turbulent weather condition (horizontal wind speed >6m/s) in late spring when the air was convergent, the BC was largely diluted down to the background level (~0.5 µg m$^{-3}$); whereas the dynamic air condition in winter the $BC_{PBL}$ was significantly elevated due to efficient transport of polluted air mass from the south. The BC showed lower surface

concentration in hot season due to much reduced BC emission from residential sector, which was consistent with previous inter-seasonal studies (Zhang et al., 2013). Both seasons exhibited similar BC mass loading in the PBL (3.5-4 µg m$^{-3}$) maybe because of the balance between the input of regional transport in late spring and the wind dilution effect in winter. The annual emission may vary, i.e. a possible reduced BC emission in 2016 compared to 2012, however the influence of emission may be of less significance by orders of

magnitude compared to the meteorological influence over Beijing region (Zheng et al., 2015). In addition, this study focused on the flights in a few specific days rather than a long-term investigation on annual variation, the influence of long-term emission variation will be of less importance.

The high $BC_{PBL}$ were mostly associated with a high thickly coated fraction ($F_{coating}$) apart from a few cases in hot season with high $BC_{PBL}$ but low $F_{coating}$, and the increase of BC coating fraction was highly modulated by

RH. This suggests when polluted days, the primary co-emitted species could significantly enhance the coatings of BC, but the secondary formation which could be enhanced by elevated RH also played important roles. The BC core size which may reflect the features of primary sources, was systematically larger in winter than in late spring, maybe due to the significant contribution of residential emission sector. However the BC core size during hot season over the NCP was still significantly larger than in Europe (McMeeking et

al., 2010;Liu et al., 2014) or North America (Schwarz et al., 2008;Metcalf et al., 2012) where the transportation emission sector dominates in hot season. This indicates the complex combined sources even in hot season in China, in line with the emission inventory that the industrial and residential sectors still dominated over the transportation sector in hot season.

The background BC, as reflected after significant removal of BC mass by turbulent weather in late spring,

showed much lower coating content and smaller core size, which suggested the coated larger BC was preferentially removed (Taylor et al., 2014). The BC in the entrainment zone ($BC_{EZ}$), which may be involved in modifying cloud microphysics, had lower coating contents than in the PBL for both seasons, but the core size showed seasonal difference. These results provide important information on constraining the direct and indirect radiative forcing impact of BC in the polluted boundary layer over the NCP.




Acknowledgments

We sincerely thank R.S. Gao in NOAA (USA) and Aijun Ding in Nanjing University (China) for their helpful discussions. This research was supported by the National key Research and Development Program of

5    China (2016YFA0602001), National Natural Science Foundation of China (41605108), Beijing Natural Science Foundation (8164057).

Data is available upon request by Delong Zhao, zhaodelong@bjmb.gov.cn.





| | Winter (PBLH=422±98m; EZ=677±144m) | | | Late Spring (PBLH=568±240m; EZ=977±330m) | | | Late Spring highly turbulent air (PBLH>1200m) | | |
|---|---|---|---|---|---|---|---|---|---|
| | BC mass (µg m⁻³) | $F_{coating}$ | Core MMD (nm) | BC mass (µg m⁻³) | $F_{coating}$ | Core MMD (nm) | BC mass (µg m⁻³) | $F_{coating}$ | Core MMD (nm) |
| **SL** | 5.22±1.78 | 0.52±0.18 | 213±6 | 3.69±0.85 | 0.41±0.27 | 196±25 | | | |
| **PBL** | 3.89±1.51 | 0.47±0.11 | 217±4 | 3.58±0.75 | 0.45±0.24 | 209±7 | 0.55±0.38 | 0.06±0.02 | 193±9 |
| **EZ** | 2.08±1.26 | 0.33±0.14 | 221±5 | 2.19±1.22 | 0.27±0.17 | 207±9 | | | |
| **FA** | 0.62±0.42 | 0.21±0.11 | 209±15 | 0.40±0.29 | 0.09±0.04 | 203±10 | | | |

Table 1. The project mean BC properties ±standard deviation over all flights in late spring and winter



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
