# Peer review of "Aircraft measurements of black carbon in the boundary layer over the North China Plain"

_Atmospheric Chemistry and Physics, 2017_

## Referee Comment (RC1) · Anonymous Referee #1 · 29 Jan 2018

This manuscript reports BC measurements on a series of aircraft flights along with some supporting met analyses. There are interesting measurements here, but the analysis presented is too unclear to merit publication in this state. I found the met analyses disjointed and unrelated to the results. The lack of comparison to prior work also makes the value of this contribution unclear. I would not recommend additional reviews without language editing as the results are very hard to follow given the language problems.

1. The language needs improvement as it contains poor grammar and obscure phrasing that prevent the reader from following the scientific points presented. Some examples are given below, but I did not provide a comprehensive list of the problems I found. I recommend that you send your manuscript to a language editing service. 2. Stop and

start of EZ need to be defined quantitatively. P.7 says "where Rib firstly reaches" but this does not provide either a quantitative or an English definition. 3. The "dynamic air condition" is not a meaningful term. 4. Why FA not FT? 5. How is coating calibrated? What is uncertainty? Why is range 120-180 nm relevant for coatings? How will missing coatings on other particles affect the results? 6. What is lower cut of sp2? Does it capture all BC? Does that bias MMD? How much? What fraction of change in MMD is noise and what fraction is variability? Quantify and show. 7. The RH of the inlet to the SP2 is not specified; were particles dry or ambient RH or somewhere in between? Lack of this information makes the coating measurement meaningless as it is an artifact of the uncontrolled drying in the inlet. Suggest removing all coating data. 8. Text says BC was bimodal but then is fit with single MMD; this is counterintuitive. Two modes should be separated and separately fit. 9. The supplement includes poorly formatted plots for each flight that lack figure and panel numbers. There is no text describing this information and it is difficult to read. 10. Fig. S7 is supposed to show agreement between hysplit and synoptic, but they are not overlaid so agreement is difficult to support. A table summarizing quantitative degree of agreement would be much more useful. 11. What does it mean to say "the BC containing larger core was favorably removed"? I could guess, but this is simply too poorly stated to merit comment.
* * *

---

## Referee Comment (RC2) · Anonymous Referee #2 · 14 Feb 2018

This paper describes airborne measurements of BC aerosol over the North China Plain near Beijing. They examine seasonal differences in concentrations, vertical distributions, size distributions and coating state and include a bit of meteorological analysis and run back trajectories. There is interesting data to present in this manuscript but the analysis should be re-worked to better support useful conclusions and the discussion includes too many statements that are pure conjecture rather than robust findings. I cannot recommend publication in ACP in its present form. In addition, significant editing is needed for English language and grammar. Below I include some comments which may help the authors re-focus and amend the analysis though this is not necessarily a comprehensive list of issues.

1. There is a lot of discussion of size distributions throughout but one is never actually

shown. I would recommend including at least a representative measured mass size distribution for BC. Also with respect to the size distributions, it does not look to me like the winter and summer PBL MMDs are significantly or systematically different. There is quite a bit of spread in the late spring values with a few flights showing markedly smaller MMDs than the others and seemingly bringing the whole average down. This probably is due to different sources as posited by the authors but I don't think there's any evidence that that is a seasonal behavior and not simply variability in transport. And transport may be more variable in the spring than in the winter but I don't really see how that is useful information. I don't think it's at all surprising that there are seasonal variations in BC sources, the question is what that means or what it can be used for.

2. All discussion of the coated fraction is rather a mess. First of all, the authors never explicitly discuss what they do with the SP2 data to derive F_coating. They often refer to it as the "evaporation time" of coatings which, if I understand what they did, is not at all the case. I believe they are looking at the time delay between the peaks in the scattering and incandescence signals in the SP2 (which is what was done in the papers referenced). This is always necessarily a bimodal distribution and the time delay has no bearing on the coating material or thickness. In the case of a bare BC particle, the peak in scattering will occur just prior to the onset of incandescence; prior to incandescence the signal is increasing because the particle is entering an ever more powerful section of the laser beam, upon incandescence the signal decreases because the particle evaporates. In the case of a coated BC particle, the scattering peak will have an initial maximum prior to the evaporation of coatings and a second maximum at the onset of incandescence. If coatings are thick enough the initial peak will be higher than the second peak and you will find a "time-delay" between the scattering and incandescence signals. In the case of thin coatings you will likely still have a local maximum at the point of coating evaporation but it will not be large enough to exceed the second scattering peak arising from the BC core and you will have no time-delay reported by the analysis software. The initial peak corresponding to coating

[Figure]

Interactive
comment

evaporation will happen at the same time relative to incandescence for every particle independent of coating material or thickness because the beam profile of the laser is very steep and the temporal resolution of the signal is not adequate to resolve tiny differences in coating evaporation rate. The exact coating thickness where the initial scattering peak will be larger than the peak at incandescence is a function of numerous factors including laser power, beam-width and BC core size so I am not convinced that the coated fractions reported for winter and spring are comparable (i.e. particles with 20nm of coating might be classified as "coated" in one phase while 40 nm of coating might be required to count as "coated" in the other phase such that a different F_coating value would be reported for identical aerosol populations depending on instrument alignment). The variations in F_coating with altitude within a given set of flights are probably usable although, again, I'm not sure that this is particularly useful information. Basically it seems like there is high variability in F_coating in the PBL, BC in the free troposphere is mostly uncoated and the entrainment zone is a mixture between the two. This is in contrast with previous findings where background BC typically has substantial coatings unless there has been a lot of wet removal (see a paper by Y. Kondo's group for example). The authors talk about "removal" but they don't make clear whether they actually saw evidence of wet removal. Can the back trajectories speak to this? Or are there nearby measurements of rainfall? I would be surprised if rain were frequent enough in both seasons that the background FT air had always recently undergone wet removal in all flights. It seems more likely that the variability in F_coating is driven by variations in sources and that the air identified as "Free troposphere" is actually also relatively recently impacted by emissions but from a different source. In the US, Europe and Japan traffic emissions are often thinly coated while biomass burning emissions are thickly coated almost immediately, perhaps what you have here is a local source dominated by biomass combustion mixing into a regional background consisting of more traffic and industrial emissions. In any case, in the absence of further evidence of what's driving these trends, I'm not sure I see the utility.

3. On a related note, with regard to the the trends in F_coating with RH: First, the authors should clarify what is happening with RH during sampling. Is the sample stream dried prior to measurement with the SP2? If not there is likely some ram heating and associated evaporation during sampling but probably not enough to fully remove all the water. Some of the F_coating increase at high RH might be due to water uptake by the coatings. If the stream was fully dried prior to sampling then it seems most likely that the more coated sources are associated with higher RH than that the high RH drives formation of coatings.

4. Did the authors calibrate the optical sizing of the SP2 for either project? Please include that in the methods and discuss the effects of variability in the laser power and alignment.

5. Considerable space is dedicated to the determination of the different layers in the profiles (i.e. PBL, entrainment zone, free troposphere) but it's not clear to me what any of the reported trends mean. I can understand separating data into PBL and free tropospheric sampling but I think the "entrainment zone" is a bit of a distraction in the absence of significantly more detail w.r.t. cloud effects and precipitation. The authors calculate and discuss "removal efficiency" defined as the difference between BC in the PBL and the EZ but it seems that most of these differences are simply dilution. And while dilution does reduce concentrations it is certainly not the same as removal. I would recommend removing this section.

6. Can the authors compare these observed loadings in the PBL and FT to any previously published results? Many parts of the discussion are lacking in context. Similarly, these measurements seem to imply that BC size distributions (and possibly coating state) from Chinese transportation sources are very different than those from the US, Europe and Japan, could the authors definitively make this case and, if so, discuss the larger implications of this finding?

In summary, I think it is valuable to show BC concentrations, size distributions and coating state from an airborne platform over China, I just don't think this particular analysis

presents the data in a very usable format. As a first outcome of this study, it would be interesting to see the variability of these BC parameters as a function of location (or back trajectory origin) and as a rough function of altitude (separated into PBL - i.e. likely reflective of local sources - and free troposphere - reflective of background conditions). Starting from those basic observations it would be interesting if the authors could comment on larger issues such as the effects on radiation, the agreement of these observations with emissions estimates and/or a regional model, the columnar load of BC, etc. As presented there are simply a lot of correlations of different BC-properties with each other and it is never clear whether these relationships are mechanistic or coincidental.

---

## Author Comment (AC1) · 18 Apr 2018

We thank both referees for their comments about the potential value of this dataset and the important suggestions on the data analysis, language editing and refocusing of the current manuscript. According to referees' suggestions, we have made thorough editing and rewriting on the previous version and now present the current version with a lot more organized and new analysis, also addressing all the concerns the referees have raised.

The major revisions we have made include:

-the mixing state of BC based on the coating evaporation time is replaced by analysis of the relative coating thickness derived from Mie lookup table using the SP2-measured scattering signal for coated BC. This analysis is further applied to analyse the BC optical properties.

-we have added a discussion of the relative humidity influence on measured BC coatings: we believe the in-situ measurement by aircraft will represent the original coating status because of the large difference of temperature outside and inside of the aircraft cabin.

-we have more explicitly presented the vertical profiles for each flight and added a few new plots to improve the contextual flow.

-we have adopted the suggestion by referee 2 and removed the discussion about BC removal. The related discussions are amended in a more appropriate manner. The exact wet removal process of BC, i.e. through cloud scavenging and subsequent precipitation is out of scope of this study and will be investigated in future work.

-the new analyses we have added are: 1) more explicit presentation of vertical profiles for all BC physical properties and meteorological analysis using new figures and tables; 2) parameterization of BC mass loading profiles; 3) the calculation of mass absorption cross section based on SP2-measured BC core and coated BC size; 4) the columnar information of BC properties including the absorption enhancement owing to coatings; 5) comparison of the SP2-constrained AAOD with the AERONET products.

The following are the answers to referees' specific comments, with the referees' original comments in blue and our answers are in normal font.

Referee 1

This manuscript reports BC measurements on a series of aircraft flights along with some supporting met analyses. There are interesting measurements here, but the analysis presented is too unclear to merit publication in this state. I found the met analyses disjointed and unrelated to the results. The lack of comparison to prior work also makes the value of this contribution unclear. I would not recommend additional re-views without language editing as the results are very hard to follow given the language problems.

We thank the referee for their comments about the value of this dataset and the important advice on improving the manuscript. We have adopted the referee's suggestions and carried out thorough editing and rewriting on the current version, including data analysis and language.

1. The language needs improvement as it contains poor grammar and obscure phrasing that prevent the reader from following the scientific points presented. Some examples are given below, but I did not provide a comprehensive list of the problems I found. I recommend that you send your manuscript to a language editing service.

We have performed thorough language editing and rewriting on the previous version.

2. Stop and start of EZ need to be defined quantitatively. P.7 says "where Rib firstly reaches" but this does not provide either a quantitative or an English definition.

We now give a more explicit definition and description on the start and end of EZ.

"In this study, the EZ is determined as the layer with $d\theta_v/dz$ >5K/km (as between the two dash lines shown in Fig. 3A), below and above the EZ both showed lower $d\theta_v/dz$ which represent the less stratified PBL and FT respectively."

3. The "dynamic air condition" is not a meaningful term.

We have amended this definition to be "southerly advection" in the revised version, which reflects the general air flow direction.

4. Why FA not FT?

Revised.

5. How is coating calibrated? What is uncertainty? Why is range 120-180 nm relevant for coatings? How will missing coatings on other particles affect the results?

Section 2.2 has been rewritten. We have also performed analysis of the coating thickness, and replaced the coating evaporation time analysis with a bulk relative coating thickness analysis for the following discussions, which is more solid and better reflects the influence of coating on BC optical properties.

6. What is lower cut of sp2? Does it capture all BC? Does that bias MMD? How much? What fraction of change in MMD is noise and what fraction is variability? Quantify and show.

The revised version gives an example of the BC core size distribution measured in the PBL during flight f20120417. The detectable BC core mass-equivalent diameter ($D_c$) is 60-500nm, below and above this range is the lower cut S/N level and saturation level of the detector respectively. The BC core mass median diameter (MMD), defined as a diameter below and above which the BC mass loading is equal. Given the majority of the BC mass was detected within this size range, the bias on MMD due to sizing range is considered to be low.

These are now added in revised section 2 and new Fig. 2.

7. The RH of the inlet to the SP2 is not specified; were particles dry or ambient RH or somewhere in between? Lack of this information makes the coating measurement meaningless as it is an artifact of the uncontrolled drying in the inlet. Suggest removing all coating data.

There is no direct drying on the inlet itself but there was dryer on the sheath flow and purge flow in the SP2 chamber. Because a constant temperature of ~25ºC was maintained inside the cabin, there was a significant temperature gradient with the outside of the cabin, especially at higher altitudes, with temperature differences of around 10-20ºC between ambient air and inside the cabin. Since increased temperature in the sampling line will enhance the saturated water vapor pressure, decreasing the RH, thus the particle will be efficiently dried during transmission through the sampling line from the inlet to the instrument. For this airborne measurement, the BC particles detected were thus warranted to be dry and the coating thickness will represent the dry state of the particle without influence by ambient RH.

We have added this discussion in the revised version.

8. Text says BC was bimodal but then is fit with single MMD; this is counterintuitive. Two modes should be separated and separately fit.

We did not observe a double mode of BC core size distribution for each flight, but what we meant was the two modes of core size distribution between the different flights. The related discussions are now revised to clarify this point.

9. The supplement includes poorly formatted plots for each flight that lack figure and panel numbers. There is no text describing this information and it is difficult to read.

We have thoroughly revised the supplement in the previous version and the figures are now more organized for the current version.

10. Fig. S7 is supposed to show agreement between hysplit and synoptic, but they are not overlaid so agreement is difficult to support. A table summarizing quantitative degree of agreement would be much more useful.

We have added this information in a revised Table 1 to summarize all information for each flight, including a more explicit presentation of backtrajectory analysis.

11. What does it mean to say "the BC containing larger core was favorably removed"? I could guess, but this is simply too poorly stated to merit comment.

We have amended all the discussions about the removal of BC in the revised version.

Referee 2

This paper describes airborne measurements of BC aerosol over the North China Plain near Beijing. They examine seasonal differences in concentrations, vertical distributions, size distributions and coating state and include a bit of meteorological analysis and run back trajectories. There is interesting data to present in this manuscript but the analysis should be re-worked to better support useful conclusions and the discussion includes too many statements that are pure conjecture rather than robust findings. I cannot recommend publication in ACP in its present form. In addition, significant editing is needed for English language and grammar. Below I include some comments which may help the authors re-focus and amend the analysis though this is not necessarily a comprehensive list of issues.

We thank the referee for their comments about the potential value of this dataset, and especially for the suggestions the referee made which helped us a lot to improve the manuscript.

1. There is a lot of discussion of size distributions throughout but one is never actually shown. I would recommend including at least a representative measured mass size distribution for BC. Also with respect to the size distributions, it does not look to me like the winter and summer PBL MMDs are significantly or systematically different. There is quite a bit of spread in the late spring values with a few flights showing markedly smaller MMDs than the others and seemingly bringing the whole average down. This probably is due to different sources as posited by the authors but I don't think there's any evidence that that is a seasonal behavior and not simply variability in transport. And transport may be more variable in the spring than in the winter but I don't really see how that is useful information. I don't think it's at all surprising that there are seasonal variations in BC sources, the question is what that means or what it can be used for.

We have adopted the referee's suggestions and included examples of BC core size distribution in the revised version to show how we calculated the mass median diameter, and also used the new Fig. 10 to show the typical cases when the shift of BC core size distribution occurred. We agree with the referee about the seasonal difference on the core MMD and have removed this part, but focused on the observed different modes of core MMD instead.

2. All discussion of the coated fraction is rather a mess. First of all, the authors never explicitly discuss what they do with the SP2 data to derive F_coating. They often refer to it as the "evaporation time" of coatings which, if I understand what they did, is not at all the case. I believe they are looking at the time delay between the peaks in the scattering and incandescence signals in the SP2 (which is what was done in the papers referenced). This is always necessarily a bimodal distribution and the time delay has no bearing on the coating material or thickness. In the case of a bare BC particle, the peak in scattering will occur just prior to the onset of incandescence; prior to incandescence the signal is increasing because the particle is entering an ever more powerful section of the laser beam, upon incandescence the signal decreases because the particle evaporates. In the case of a coated BC particle, the scattering peak will have an initial maximum prior to the evaporation of coatings and a second maximum at the onset of incandescence. If coatings are thick enough the initial peak will be higher than the second peak and you will find a "time-delay" between the scattering and incandescence signals. In the case of thin coatings you will likely still have a local maximum at the point of coating evaporation but it will not be large enough to exceed the second scattering peak arising from the BC core and you will have no time-delay reported by the analysis software. The initial peak corresponding to coating evaporation will happen at the same time relative to incandescence for every particle independent of coating material or thickness because the beam profile of the laser is very steep and the temporal resolution of the signal is not adequate to resolve tiny differences in coating evaporation rate. The exact coating thickness where the initial scattering peak will be larger than the peak at incandescence is a function of numerous factors including laser power, beam-width and BC core size so I am not convinced that the coated fractions reported for winter and spring are comparable (i.e. particles with 20nm of coating might be classified as "coated" in one phase while 40 nm of coating might be required to count as "coated" in the other phase such that a different F_coating value would be reported for identical aerosol populations depending on instrument alignment). The variations in F_coating with altitude within a given set of flights are probably usable although, again, I'm not sure that this is particularly useful information.

We thank the referee for their comprehensive explanation on the disadvantage of the $F_{coating}$ analysis and now we have replaced the $F_{coating}$ analysis with a bulk relative coating thickness method which will better reflect the true mixing state of BC. This calculation is largely independent of the uncertainties owing to the smaller particles, given their lower contribution to the integrated volume.

Basically it seems like there is high variability in F_coating in the PBL, BC in the free troposphere is mostly uncoated and the entrainment zone is a mixture between the two. This is in contrast with previous findings where background BC typically has substantial coatings unless there has been a lot of wet removal (see a paper by Y. Kondo's group for example). The authors talk about "removal" but they don't make clear whether they actually saw evidence of wet removal. Can the back trajectories speak to this? Or are there nearby measurements of rainfall? I would be surprised if rain were frequent enough in both seasons that the background FT air had always recently undergone wet removal in all flights. It seems more likely that the variability in F_coating is driven by variations in sources and that the air identified as "Free troposphere" is actually also relatively recently impacted by emissions but from a different source. In the US, Europe and Japan traffic emissions are often thinly coated while biomass burning emissions are thickly coated almost immediately, perhaps what you have here is a local source dominated by biomass combustion mixing into a regional background consisting of more traffic and industrial emissions. In any case, in the absence of further evidence of what's driving these trends, I'm not sure I see the utility.

We have performed both $F_{coating}$ and bulk relative coating thickness analysis on the BC mixing state and both methods showed consistent results, therefore we believe the results presented here are true. In the revised version, we have more explicitly presented the backtrajectory analysis.

Given the measurements were conducted over the very polluted urban environment, the sources from different sectors were found to be well mixed, and we found no obvious relationship between airmass direction and physical properties of BC in either the PBL or FT. According to referee's suggestion, we have included discussions about source-specific information of BC in the revised version, comparing with other studies in the literature and explained the possible reason why in the FT the BC had a reduced coating. In addition, we have added the references the referee mentioned and made related discussions on the possible removal process of BC.

3. On a related note, with regard to the trends in F_coating with RH: First, the au-thors should clarify what is happening with RH during sampling. Is the sample stream dried prior to measurement with the SP2? If not there is likely some ram heating and associated evaporation during sampling but probably not enough to fully remove all the water. Some of the F_coating increase at high RH might be due to water uptake by the coatings. If the stream was fully dried prior to sampling then it seems most likely that the more coated sources are associated with higher RH than that the high RH drives formation of coatings.

There is no direct drying on the inlet itself but there was dryer on the sheath flow and purge flow in the SP2 chamber. Because a constant temperature of ~25ºC was maintained inside the cabin, there was a significant temperature gradient with the outside of the cabin, especially at higher altitudes, with temperature differences of around 10-20ºC between ambient air and inside the cabin. Since increased temperature in the sampling line will enhance the saturated water vapor pressure, decreasing the RH, thus the particle will be efficiently dried during transmission through the sampling line from the inlet to the instrument. For this airborne measurement, the BC particles detected were thus warranted to be dry and the coating thickness will represent the dry state of the particle without influence by ambient RH.

We have added this discussion in the revised version.

4. Did the authors calibrate the optical sizing of the SP2 for either project? Please include that in the methods and discuss the effects of variability in the laser power and alignment.

The laser optics (the YAG crystal and mirror position) and aerosol jet position relative to the detecting chamber were optimally aligned before flight. The laser power of the SP2 was calibrated and monitored using mono-dispersed polystyrene latex spheres (PSL) on a weekly basis, and the variation of peak intensity of the PSL at a given size which reflects the laser power maintained within ±6% among flights. We have added this information in the revised version.

5. Considerable space is dedicated to the determination of the different layers in the profiles (i.e. PBL, entrainment zone, free troposphere) but it's not clear to me what any of the reported trends mean. I can understand separating data into PBL and free tropospheric sampling but I think the "entrainment zone" is a bit of a distraction in the absence of significantly more detail w.r.t. cloud effects and precipitation. The authors calculate and discuss "removal efficiency" defined as the difference between BC in the PBL and the EZ but it seems that most of these differences are simply dilution. And while dilution does reduce concentrations it is certainly not the same as removal. I would recommend removing this section.

We have amended the discussion on the removal of BC in the EZ. Per referee's suggestion, we have removed the previous section regarding the different characteristics of BC in the EZ.

6. Can the authors compare these observed loadings in the PBL and FT to any previously published results? Many parts of the discussion are lacking in context. Similarly, these measurements seem to imply that BC size distributions (and possibly coating state) from Chinese transportation sources are very different than those from the US, Europe and Japan,

could the authors definitively make this case and, if so, discuss the larger implications of this finding?

We have thoroughly edited and rewritten the sections the referee mentioned and comparisons with the existing literatures have been included in the revised version.

In summary, I think it is valuable to show BC concentrations, size distributions and coating state from an airborne platform over China, I just don't think this particular analysis presents the data in a very usable format. As a first outcome of this study, it would be interesting to see the variability of these BC parameters as a function of location (or back trajectory origin) and as a rough function of altitude (separated into PBL - i.e. likely reflective of local sources - and free troposphere - reflective of background conditions). Starting from those basic observations it would be interesting if the authors could comment on larger issues such as the effects on radiation, the agreement of these observations with emissions estimates and/or a regional model, the columnar load of BC, etc. As presented there are simply a lot of correlations of different BC-properties with each other and it is never clear whether these relationships are mechanistic or coincidental.

We appreciate referee's important suggestions, which have greatly helped to improve the previous version. We have added a few new points to present the data in a more usable way. The new points added include:

-the calculation of mass absorption cross section of BC based on the SP2 measured BC core size and coating thickness;

-a more explicit presentation and explanation of the layer-segregated information of BC physical properties;

-columnar information of BC mass loading, absorption, and comparison with the SP2-constrained AAOD with AERONET AAOD product.

Regarding the point about comparing the results with regional model, we are planning future work on the model-measurement comparison based on this dataset.